# Advancing African-Accented English Speech Recognition: Epistemic Uncertainty-Driven Data Selection for Generalizable ASR Models

## Abstract

Accents play a pivotal role in shaping human communication, enhancing our ability to convey and comprehend messages with clarity and cultural nuance. While there has been significant progress in Automatic Speech Recognition (ASR), African-accented English ASR has been understudied due to a lack of training datasets, which are often expensive to create and demand colossal human labor. Combining several active learning paradigms and the core-set approach, we propose a new multi-rounds adaptation process that uses epistemic uncertainty to automate the annotation process, significantly reducing the associated costs and human labor. This novel method streamlines data annotation and strategically selects data samples contributing most to model uncertainty, enhancing training efficiency. We define a new U-WER metric to track model adaptation to hard accents. We evaluate our approach across several domains, datasets, and high-performing speech models. Our results show that our approach leads to a 27% WER relative average improvement while requiring, on average, 45% less data than established baselines. Our approach also improves out-of-distribution generalization for very low-resource accents, demonstrating its viability for building generalizable ASR models in the context of accented African ASR. We open-source the code here.

## 1 Introduction

Automatic Speech Recognition (ASR) is an active research area that powers voice assistant systems (VASs) like Siri and Cortana, enhancing daily communication (Kodish-Wachs et al. (2018); Finley et al. (2018); Zapata & Kirkedal (2015)). Despite this progress, no current VASs include African languages, which account for about 31% of the world languages, and their unique accents (Eberhard et al. (2019); Tsvetkov (2017)). This gap underscores the need for ASR systems that can handle the linguistic diversity and complexity of African languages, especially in crucial applications like healthcare. Due to the lack of representations of these languages and accents in training data, existing ASR systems often perform inadequately, even mispronouncing African names (Olatunji et al. (2023a)).

To address these challenges, our work focuses on adapting pretrained speech models to better transcribe African-accented English, defined by unique intonations and pronunciations (Benzeghiba et al. (2007); Hinsvark et al. (2021)). We use **epistemic uncertainty (EU)** (Kendall & Gal (2017)) to guide the adaptation process by identifying gaps in model knowledge and prioritizing data for the model to learn from next. This is particularly beneficial in scenarios where data annotation is costly or time-consuming, as often seen in the African context (Badenhorst & De Wet (2019; 2017); Barnard et al. (2009); Yemmene & Besacier (2019); DiChristofano et al. (2022); Dossou et al. (2022); Dossou & Emezue (2021)). EU also improves robustness and encourages exploration to mitigate inductive bias from underrepresented accents. Common approaches to compute EU include Monte Carlo Dropout (MC-Dropout) (Gal & Ghahramani (2016)) and Deep Ensembles (Lakshminarayanan et al. (2017)), with the latter being more effective but computationally expensive. Due to resource constraints, we use MC-Dropout, which requires models to have dropout components during pretraining.

To further enhance the efficiency and effectiveness of model adaptation, we employ **Active Learning (AL)** techniques. AL leverages epistemic uncertainty to select the most informative data points from an unlabeled dataset for labeling, thereby improving model performance with fewer training instances. Common types of AL include Deep Bayesian Active Learning (DBAL) (Gal et al. (2017); Houlsby et al. (2011)) and Adversarial Active Learning (AAL) (Ducoffe & Precioso (2018)). AAL selects examples likely to be misclassified by the current model, refining it iteratively by challenging it with complex cases to enhance robustness. The core-set approach (CSA) (Sener & Savarese (2017)) is also related, as it selects a subset of training data to ensure a model trained on this subset performs comparably to one trained on the entire dataset, addressing scalability and efficiency. A critical component of AL is the **acquisition function (AF)**, which determines the most informative samples from an unlabeled dataset for labeling. Key AFs include uncertainty sampling (US) (Liu & Li (2023)), Bayesian Active Learning by Disagreement (BALD) (Gal et al. (2017)), and BatchBALD (Kirsch et al. (2019)). US targets data points with the highest model uncertainty. BALD maximizes the mutual information between model parameters and predictions. BatchBALD is an extension of BALD that selects multiple samples simultaneously but may choose redundant points. US is the least computationally expensive, making it ideal for efficient data labeling.

In this work, we leverage and combine DBAL, AAL, US, and CSA in the following way (in order): First, we integrate the CSA by leveraging smaller training subsets ($\sim 45\%$ smaller than the full available training sets). Second, we use DBAL with MC-Dropout, to apply dropout during training and inference to estimate Bayesian posterior distribution. This allows us to practically and efficiently estimate EU in the models used (Gal et al. (2017)) (see section 3.2 for more details). Third, we use the estimated EU and integrate the idea of AAL by using the US acquisition function.

We evaluate our approach across **several domains** (general, clinical, general+clinical aka *both*), **several datasets** (AfriSpeech-200 (Olatunji et al. (2023b)), SautiDB (Afonja et al. (2021b)), MedicalSpeech, CommonVoices English Accented Dataset (Ardila et al. (2019))), and **several high-performing speech models** (Wav2Vec2-XLSR-53 (Conneau et al. (2020)), HuBERT-Large (Hsu et al. (2021)), WavLM-Large (Chen et al. (2022)), and NVIDIA Conformer-CTC Large (en-US) (Gulati et al. (2020))). **Our results show a 27% Word Error Rate (WER) relative average improvement while requiring on average 45% less data than established baselines.** We also adapt the standard WER to create a new metric called Uncertainty WER (U-WER) to track model adaptation to African accents.

The impact of our approach is substantial. It develops more robust, generalizable, and cost-efficient African-accented English ASR models and reduces dependency on large labeled datasets, enabling deployment in various real-world scenarios. Our results show improved generalization for out-of-distribution (OOD) cases, especially for accents with minimal resources, addressing specific challenges in African-accented ASR. Additionally, by focusing on equitable representation in ASR training, our methodology promotes fairness in AI, ensuring technology serves users across diverse linguistic backgrounds without bias (Selbst et al. (2019); Mitchell et al. (2019); Mehrabi et al. (2021)). Our contributions are listed as follows:

- we combine DBAL, AAL, CSA, and EU to propose a novel way to adapt several high-performing pretrained speech models to build efficient African-accented English ASR models,
- we evaluate our approach across several speech domains (clinical, general, *both*), and African-accented speech datasets (AfriSpeech-200 (Olatunji et al. (2023b)), SautiDB (Afonja et al. (2021b)), MedicalSpeech and CommonVoices English Accented Dataset (Ardila et al. (2019))), while providing domain and accent-specific analyses,
- we define a new and simple metric called U-WER that allows us to measure and track how the variance of the model, across hard accents, changes over the adaptation process,
- we show that our approach improves the relative average WER performance by 27% while significantly reducing the required amount of labeled data (by $\sim$45%),
- we show, based on additional AL experiments, that our approach is also efficient in real-world settings where there are no gold transcriptions.

## 2 BACKGROUND AND RELATED WORKS

### 2.1 CHALLENGES FOR AFRICAN-ACCENTED ASR

State-of-the-art (SOTA) ASR technologies, powered by deep learning and neural network architectures like transformers, achieve high accuracy with Standard American English and major European languages. However, they often fail with African accents due to high variability in pronunciation and lack of quality speech data (Koenecke et al. (2020); Das et al. (2021)). This results in racial bias, poor performance, and potential social exclusion as speakers might alter their speech to be understood (Koenecke et al. (2020); Koenecke (2021); Chiu et al. (2018); Mengesha et al. (2021)). Enhancing ASR for African languages is crucial for equitable voice recognition, especially in healthcare, education, and customer service. Solutions should focus on diversifying training datasets and developing robust modeling techniques tailored to the unique characteristics of these languages.

### 2.2 ACTIVE LEARNING

AL aims to reduce the number of labeled training examples by automatically processing the unlabeled examples and selecting the most informative ones concerning a given cost function for a human to label. It is particularly effective when labeled data is scarce or expensive, optimizing the learning process by focusing on samples that most improve the model performance and generalization (Settles (2009); Gal et al. (2017)). Several works have demonstrated its effectiveness and efficiency. An AL setup involves an unlabeled dataset $\mathcal{D}_{\text{pool}} = \{\mathbf{x}_i\}_{i=1}^{n_{\text{pool}}}$, a labeled training set $\mathcal{D}_{\text{train}} = \{\mathbf{x}_i, y_i\}_{i=1}^{n_{\text{train}}}$, and a predictive model with likelihood $p_w(y|x)$ parameterized by $w \sim p(W|\mathcal{D}_{\text{train}})$ ($W$ are the parameters of the model). The setup assumes the presence of an oracle to provide predictions y for all $x_i \in \mathcal{D}_{\text{pool}}$. After training, a batch of data $\{\mathbf{x}_i^*\}_{i=1}^{b}$ is selected from $\mathcal{D}_{\text{pool}}$ based on its EU.

In (Hakkani-Tür et al. (2002)), AL was applied to a toy dataset of *How May I Help You* recordings. Confidence scores were estimated for each word and used to compute the overall confidence score for the audio sample. This approach achieved competitive results using 27% less data compared to the baseline. In (Riccardi & Hakkani-Tur (2005)), the authors estimated confidence scores for each utterance using an online algorithm with the lattice output of a speech recognizer. The utterance scores were filtered through an informativeness function to select an optimal subset of training samples, reducing the labeled data needed for a given WER by over 60%. (Nallasamy et al. (2012)) experimented with AL for accent adaptation in speech recognition. They adapted a source recognizer to the target accent by selecting a small, matched subset of utterances from a large, untranscribed, multi-accented corpus for human transcription. They used a cross-entropy-based relevance measure alongside uncertainty-based sampling. However, their experiments on Arabic and English accents showed worse performance compared to baselines while using more hours of recordings.

## 3 DATASETS AND METHODOLOGY

### 3.1 DATASETS

We used the AfriSpeech-200 dataset (Olatunji et al. (2023b)), a 200-hour African-accented English speech corpus for clinical and general ASR. This dataset includes 120+ African accents from five language families: Afro-Asiatic, Indo-European, Khoe-Kwadi (Hainum), Niger-Congo, and Nilo-Saharan, representing African regional diversity. It was crowd-sourced from over 2000 African speakers from 13 anglophone countries in sub-Saharan Africa and the US (see Table 1).

To demonstrate the dataset-agnostic nature of our approach, we also explored three additional datasets: (1) **SautiDB** (Afonja et al. (2021a)), Nigerian accent recordings with 919 audio samples at a 48kHz sampling rate, totaling 59 minutes; (2) **MedicalSpeech**[1], containing 6,661 audio utterances of common medical symptoms, totaling 8 hours; and (3) **CommonVoices English Accented Dataset**, a subset of English Common Voice (version 10) (Ardila et al. (2019)), excluding western accents to focus on low-resource settings.

---

[1] https://www.kaggle.com/datasets/paultimothymooney/medical-speech-transcription-and-intent

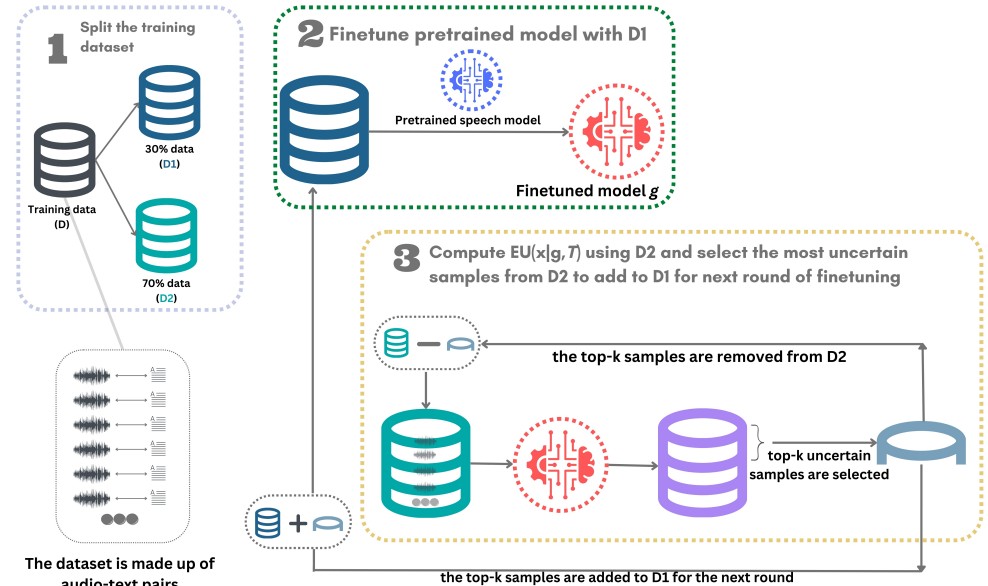

Figure 1: Our adaptation pipeline involves several phases. Initially, the dataset is split into a training set ($D1 = \mathcal{D}^*_{\text{train}}$, 30%) and a pool dataset ($D2 = \mathcal{D}_{\text{pool}}$, 70%). In the iterative process between phases 2 and 3, $D1$ is used to finetune a pretrained model. The top-k samples are selected using defined strategies and added to $D1$ for the next round. For more details on the uncertainty selection strategy, see section 3.2.

Table 1: AfriSpeech-200 Dataset statistics

| AfriSpeech Dataset Statistics | |
|---|---|
| Total duration | 200.91 hrs |
| Total clips | 67,577 |
| Unique Speakers | 2,463 |
| Average Audio duration | 10.7 seconds |
| **Speaker Gender Ratios - # Clip %** | |
| Female | 57.11% |
| Male | 42.41% |
| Other/Unknown | 0.48% |
| **Speaker Age Groups - # Clips** | |
| <18yrs | 1,264 (1.88%) |
| 19-25 | 36,728 (54.58%) |
| 26-40 | 18,366 (27.29%) |
| 41-55 | 10,374 (15.42%) |
| >56yrs | 563 (0.84%) |
| **Clip Domain - # Clips** | |
| Clinical | 41,765 (61.80%) |
| General | 25,812 (38.20%) |

## 3.2 METHODOLOGY

In our approach, to compute EU for a given input $x \in \mathcal{D}_{\text{pool}}$, we perform MC-Dropout to obtain multiple stochastic forward passes through a finetuned ASR model $g$ with likelihood $p_{w \sim p(\mathbf{W}|\mathcal{D}^*_{\text{train}})}(y|x)$ where $\mathbf{W}$ is the weights of $g$. Let $f$ be a function that computes the WER between the predicted and the target transcripts. Let $T$ be the number of stochastic forward passes. For each pass $t$, we apply dropout, obtain the output transcript, and compute the WER:

$$f_t = f(y, \hat{y}_t); \hat{y}_t = g(\mathbf{W}, \tilde{x}_t); \tilde{x}_t = x \cdot \mathbf{M}_t$$

Table 2: Dataset splits showing speakers, number of clips, and speech duration in Train/Dev/Test splits.

| AfriSpeech-200 Dataset Splits | | | | |
|---|---|---|---|---|
| **Item** | **Train ($\mathcal{D}^*_{\text{train}}$)** | **Dev** | **Test** | **AL Top-*k*** |
| # Speakers | 1466 | 247 | 750 | ✗ |
| # Hours | 173.4 | 8.74 | 18.77 | ✗ |
| # Accents | 71 | 45 | 108 | ✗ |
| Avg secs/speaker | 425.81 | 127.32 | 90.08 | ✗ |
| clips/speaker | 39.56 | 13.08 | 8.46 | ✗ |
| speakers/accent | 20.65 | 5.49 | 6.94 | ✗ |
| secs/accent | 8791.96 | 698.82 | 625.55 | ✗ |
| # general domain | 21682 (*6504) | 1407 | 2723 | 2000 |
| # clinical domain | 36318 (*10895) | 1824 | 3623 | 3500 |
| # *both* domain | 58000 (*17400) | 3221 | 6346 | 6500 |

---

**Algorithm 1** Selection of the best-generated transcript in Active Learning for an input Sample $x$

---

1: we generate the predictions $\hat{y}_1, .., \hat{y}_T$ corresponding to each stochastic forward pass ($T$=10 in our experiments)
2: we define a list variable called wer_list and a dictionary variable called wer_target_dict, respectively tracking all pairwise WERs and the average pairwise WER of each target prediction
3: **for** $\forall$ i,j $\in \{1, ..., T\}$ **do**
4: $\quad \rightarrow \hat{y}_i$ is set as target transcription
5: $\quad \rightarrow$ target_wer = list()
6: $\quad$ **for** for j $\neq$ i **do**
7: $\quad\quad w = WER(\hat{y}_j, \hat{y}_i)$
8: $\quad\quad$ wer_list.append($w$)
9: $\quad\quad$ target_wer.append($w$)
10: $\quad$ **end for**
11: $\quad wer_{\hat{y}_i}$ = mean(target_wer)
12: $\quad$ wer_target_dict[$\hat{y}_i$] $\leftarrow wer_{\hat{y}_i}$
13: **end for**
14: $\hat{y}_{best} = \hat{y}_i$, such that wer_target_dict[$\hat{y}_i$] = min(wer_target_dict.values())
15: **return** ($p_{best}$, std(wer_list))

---

where $\mathbf{M}_t$ is a binary mask matrix sampled independently for each pass. EU($x|g, T$) can then be estimated from the $T$ stochastic forward passes as follows:

$$\text{EU}(x|g,T) = \sigma(f) = \sqrt{\frac{1}{T}\sum_{t=1}^{T} f_t^2 - \left(\frac{1}{T}\sum_{t=1}^{T} f_t\right)^2} \tag{1}$$

The use of MC-Dropout requires models to have dropout components during training. This excludes some models like Whisper (Radford et al. (2022)), which we still finetuned and evaluated as a baseline. We use four state-of-the-art pretrained models: Wav2Vec2-XLSR-53, HuBERT-Large, WavLM-Large, and NVIDIA Conformer-CTC Large (en-US), referred to as Wav2Vec, Hubert, WavLM, and Nemo, respectively.

### 3.2.1 UNCERTAINTY WER

To handle diverse accents, we aim to reduce the EU of the models across hard accents after each adaptation round. We define a metric called *U-WER* to track this. To compute U-WER($a$) where $a$ is a hard accent, we condition EU on $a$:

$$\text{EU}(x|g,T,a) = \sigma(f_a) = \sqrt{\frac{1}{T}\sum_{t=1}^{T} f_{t,a}^2 - \left(\frac{1}{T}\sum_{t=1}^{T} f_{t,a}\right)^2} \tag{2}$$

where $x_a$ is the audio sample with accent $a$ and

$$f_{t,a} = f(y_a, \hat{y}_{t,a}); \hat{y}_{t,a} = g(\mathbf{W}, \tilde{x}_{t,a}); \tilde{x}_{t,a} = x_a \cdot \mathbf{M}_t$$

Ideally, U-WER→0. The rationale behind U-WER is that as beneficial data points are acquired, U-WER should decrease or remain constant, indicating increased robustness, knowledge, and performance, which is crucial for generalization. During AL, U-WER is computed using pairwise WER scores among predicted transcriptions, not gold transcriptions (see section 3.3). To select the best-generated transcript for unlabeled speech $x$, we follow Algorithm 1.

---

**Algorithm 2** Adaptation Round using Epistemic Uncertainty-based Selection

---

**Require:** Pretrained Model $\mathcal{M}$, Training Dataset $\mathcal{D}^*_{\text{train}}$, Validation Dataset $\mathcal{D}_{Val}$, and Pool Dataset $\mathcal{D}_{\text{pool}}$

1: $\mathcal{N} \leftarrow 3$              ▷ Number of Adaptation Rounds
2: $T \leftarrow 10$            ▷ Number of Stochastic Forward Passes
3: **for** $k \leftarrow 1$ to $\mathcal{N}$ **do**
4:   $g \leftarrow$ Finetune $\mathcal{M}$ on $\mathcal{D}^*_{\text{train}}$ using $\mathcal{D}_{Val}$
5:   $\mathcal{EUL} \leftarrow \{\}$           ▷ List of Uncertainty Scores
6:   **for** $x$ in $\mathcal{D}_{\text{pool}}$ **do**         ▷ x is an audio sample
7:     $\text{EU}_x \leftarrow \text{EU}(x|g, T)$      ▷ Epistemic Uncertainty of $x$
8:     $\mathcal{EUL} \leftarrow \mathcal{EUL} \cup \{(x, \text{EU}_x)\}$
9:   **end for**
10:   $topk \leftarrow \{x_1, ..., x_k\}$        ▷ Samples with highest $\mathcal{EU}$
11:   $\mathcal{D}^*_{\text{train}} \leftarrow \mathcal{D}^*_{\text{train}} \cup topk$
12:   $\mathcal{D}_{\text{pool}} \leftarrow \mathcal{D}_{\text{pool}} \setminus topk$
13: **end for**

---

### 3.3 EXPERIMENTAL DESIGN

To work within our framework, we define the following selection strategies:

- **random**: Randomly selects audio samples from $\mathcal{D}_{\text{pool}}$.
- **EU-Most**: Selects the most uncertain audio samples from $\mathcal{D}_{\text{pool}}$ to add to $\mathcal{D}_{\text{train}}$.
- **AL-EU-Most**: Combines AL with the **EU-Most** strategy to finetune the pretrained model.

We also define **standard fine-tuning (SFT)** as baseline using all available data for finetuning. In SFT, $\mathcal{D}_{\text{pool}}$ is empty. While running the defined strategies in our framework, we **impose data constraints, not exceeding 60-65% of the initial dataset after all adaptation rounds.** $\mathcal{D}^*_{\text{train}}$ is 30% of $\mathcal{D}_{\text{train}}$, and $\mathcal{D}_{\text{pool}}$ is 70% of $\mathcal{D}_{\text{train}}$. This simulates realistic scenarios where not all data might be available, testing the approach's robustness and efficiency under constraints. The number of samples in $\mathcal{D}_{\text{train}}$ and $\mathcal{D}_{\text{pool}}$ is based on available training examples for each domain (see Tables 2, 4, and Appendix A.1).

Our EU-based pipeline is shown in Figure 1 and Algorithm 2. In each adaptation round, we use a finetuned model and a selection strategy to choose samples from $\mathcal{D}_{\text{pool}}$ to add to $\mathcal{D}^*_{\text{train}}$. During AL experiments, we consider samples from $\mathcal{D}_{\text{pool}}$ as unlabeled: (1) using MC-Dropout, we obtain $n = 10$ different input representations per audio sample to get $n$ different transcripts; (2) we then learn to select the best-generated transcription as the target transcription according to Algorithm 1.

Our experiments aim to answer the following research questions:

1. how does the pretrained ASR model adapt to a set of African accents across adaptation rounds and domains?
2. which selection strategy (**EU-most** or **random**) works better, and for which domain(s)?
3. which domain(s) help the model perform better, and how does the model perform (in terms of uncertainty) across the domain(s)?
4. what is the impact of EU-based selection on the model's efficiency in low-resource data scenarios?
5. is uncertainty-based selection, model, and dataset agnostic?

U-WER will answer question 4. To answer question 5, we evaluated our approach with three additional pretrained models (Nemo, WavLM, and Hubert) and across three external datasets (SautiDB, CommonVoices English Accented Dataset, and MedicalSpeech). For consistency and better visualization, we considered the top-10 (in terms of frequency) accents across three adaptation rounds and

Table 3: We used Wav2Vec to conduct initial experiments across domains and strategies to identify the best selection strategy. Models marked with ** are used to demonstrate that our algorithm is model agnostic, utilizing the **EU-Most** selection strategy, which has been proven the most effective. Our AL experiments also use this strategy. Wav2Vec, using the **random** strategy, scored 0.1111, 0.3571, and 0.1666 for the general, clinical, and *both* domains, respectively. We omit **random** results to enhance readability.

| Model | General | | | Clinical | | | Both | | |
|---|---|---|---|---|---|---|---|---|---|
| | Baseline | EU-Most | AL-EU-Most | Baseline | EU-Most | AL-EU-Most | Baseline | EU-Most | AL-EU-Most |
| Wav2vec | 0.2360 Olatunji et al. (2023b) | **0.1011** | 0.1059 | 0.3080 Olatunji et al. (2023b) | **0.2457** | 0.2545 | 0.2950 Olatunji et al. (2023b) | 0.1266 | **0.1309** |
| **Hubert | **0.1743** | 0.1901 | 0.1887 | 0.2907 | **0.2594** | 0.2709 | **0.2365** | 0.2453 | 0.2586 |
| **WavLM | 0.1635 | **0.1576** | 0.1764 | 0.3076 | **0.2313** | 0.2537 | 0.2047 | **0.1897** | 0.1976 |
| **Nemo | 0.2824 | **0.1765** | 0.1815 | 0.2600 | **0.2492** | 0.2526 | 0.3765 | **0.2576** | 0.2610 |
| Average Performance | 0.2141 | **0.1563** | 0.1631 | 0.2916 | **0.2464** | 0.2579 | 0.2782 | **0.2043** | 0.2120 |
| Whisper-Medium | 0.2806 | - | - | 0.3443 | - | - | 0.3116 | - | - |

both selection strategies to answer questions 1-4. For very low-resource settings, we considered the five accents with the least recording hours.

For our experiments, we used 6 RTX8000 GPUs and 4 A100 GPUs. Training and evaluation were conducted over a month. Our models have approximately 311 million trainable parameters. Each audio sample was normalized and processed at a 16kHz sample rate. We used default parameters from the HuggingFace library for each pretrained model.

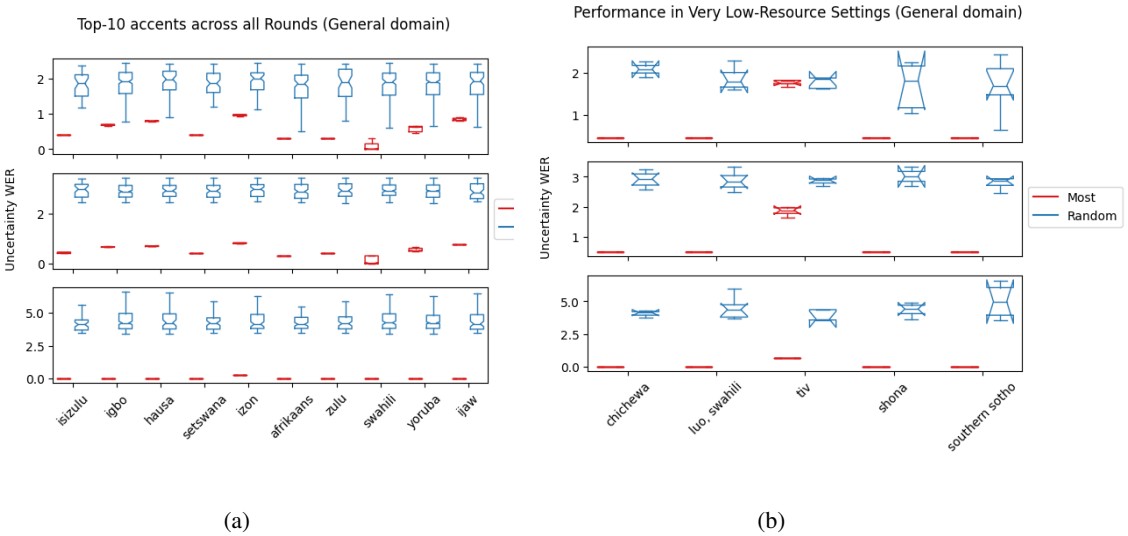

(a)                 (b)

Figure 2: WER Performance on Accents from General Domain

Table 4: WER Evaluation Results on External Datasets, with $\alpha \in [0.60, 0.65]$ as described in Section 3.1 and on Figure 1. We see an improvement for WER using our approach in all datasets, showing that our algorithm is dataset-agnostic.

| Dataset | Split and Size for our approach | | | | Finetuning Epochs | Baseline | EU-Most |
|---|---|---|---|---|---|---|---|
| | $\mathcal{D}^*_{\text{train}}$ | $\mathcal{D}_{\text{pool}}$ | Top-$k$ | Test | | ($\mathcal{D}_{\text{train}}$) | ($\mathcal{D}^*_{\text{train}} + \alpha\mathcal{D}_{\text{pool}}$) |
| SautiDB Afonja et al. (2021a) | 234 | 547 | 92 | 138 | 50 | 0.50 | **0.12** |
| MedicalSpeech | 1598 | 3730 | 1333 | 622 | 5 | 0.30 | **0.28** |
| CommonVoices English Accented Dataset (v10.0) Ardila et al. (2019) | 26614 | 62100 | 10350 | 232 | 5 | 0.50 | **0.22** |
| Average | ✗ | ✗ | ✗ | ✗ | ✗ | 0.43 | **0.20** |

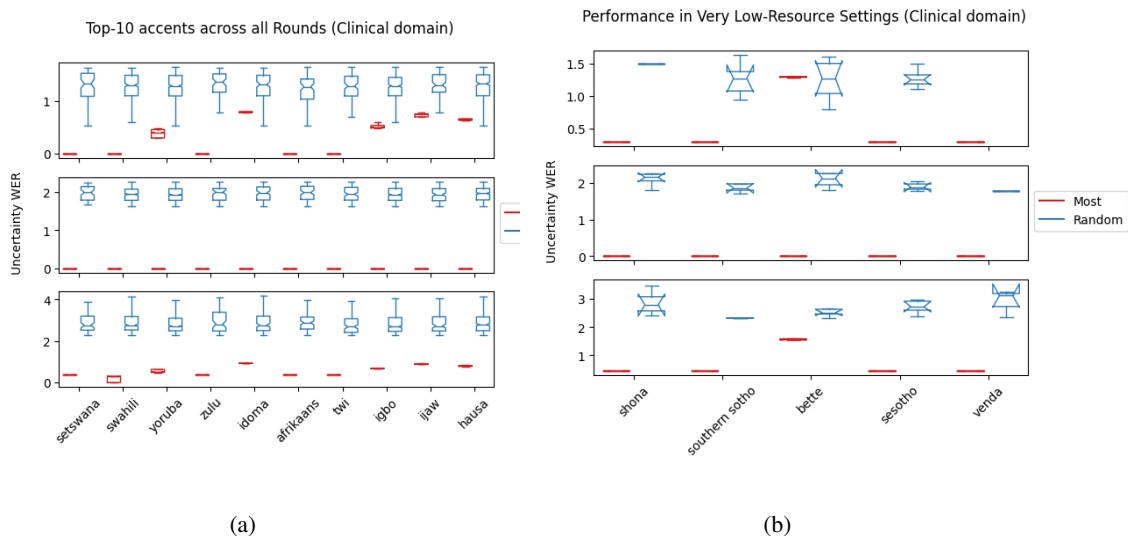

(a)                                                        (b)

Figure 3: WER Performance on Accents from Clinical Domain

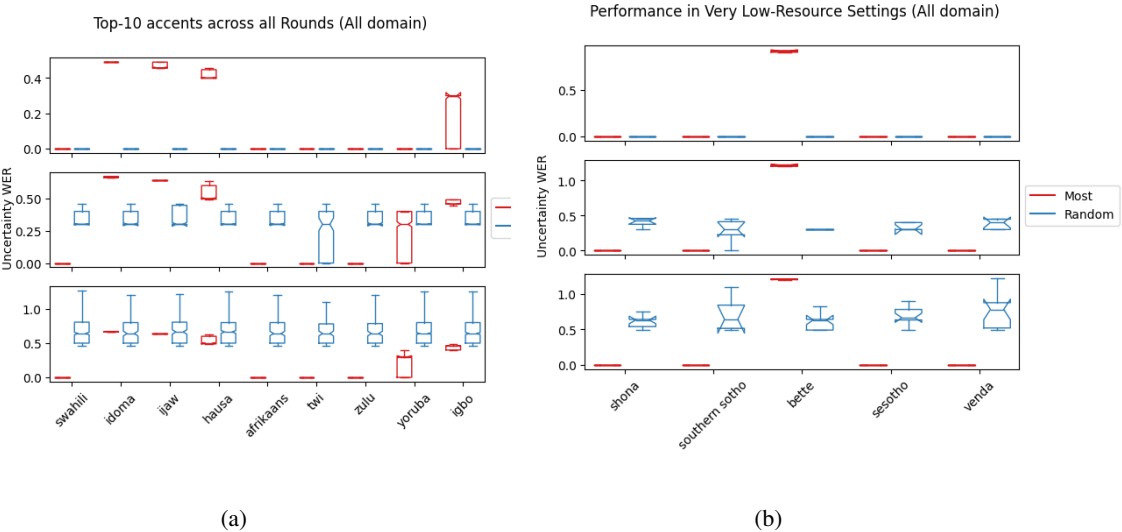

(a)                                                        (b)

Figure 4: WER Performance on Accents from Clinical+General (*Both*) Domain

## 4 RESULTS AND DISCUSSION

To assess the performance improvement for each domain, we compute the relative average improvement

$$\text{RIA}_{wer,d} = \left( \frac{b_{wer}^d - s_{wer}^d}{b_{wer}^d} \right) \times 100\%$$

where $b_{wer}^d$ and $s_{wer}^d$ are the average WER respectively of the baseline, and the best selection strategy, in a domain $d \in \{general, clinical, both\}$. A higher percentage reflects a higher improvement in our approach.

Table 3 shows the results of our experiments, indicating that our uncertainty-based selection approach significantly outperforms the baselines across **all models, domains, and datasets: general (27.00%), clinical (15.51%), and *both* (26.56%)**. Our approach also surpasses Whisper-Medium (Olatunji

et al. (2023b); Radford et al. (2023)), demonstrating the importance of epistemic uncertainty in ASR for low-resource languages. The **EU-Most** selection strategy proves to be the most effective across all domains due to the model's exposure to highly uncertain samples, enhancing robustness and performance. However, performance disparities between general and clinical domains are noted, likely due to clinical samples complexity. These findings confirm **EU-Most** as the superior selection strategy, as detailed in the results and illustrated in Figures 2, 3, and 4. This answers question 2.

To identify the best learning signals within a diverse dataset characterized by various accents, speaker traits, genders, and ages, we analyzed the top-k uncertain accents using the **EU-Most** selection strategy. Our findings, illustrated in Figures 2, 3, and 4, show that the top-10 accents (most represented in recording hours) remained consistently challenging across all rounds of analysis (refer to Figures 2, 3, 4 and Tables 6, 7, and 8). These accents, characterized by high linguistic richness and variability, aid in model learning and enhance performance over time. We positively answer questions 1 and 3, confirming that the model adapts effectively to the beneficial accents from all domains. This demonstrates that the model adapts qualitatively and quantitatively well to the beneficial accents and benefits from all domains. Figures 2 (b), 3 (b), and 4 (b) also affirm positive outcomes for question 4, showing consistent improvement or stable performance on low-resource accents. This highlights the relevance of our approach in addressing the challenges associated with the low resource availability typical of many African accents and languages.

To demonstrate the agnostic aspect of our approach, we evaluated it with three additional pretrained models (Hubert, WavLM, and Nemo) and three datasets containing accented speech in general and clinical domains, using only the **EU-Most** selection strategy. The results, shown in Tables 3 and 4, indicate that our uncertainty-based adaptation approach consistently outperforms baselines. This confirms that our approach applies to any model architecture and dataset and allows us to answer positively question 5.

## 5 CONCLUSION

We combined several AL paradigms, the CSA, and the EU to create a novel multi-round adaptation process for high-performing pretrained speech models, aiming to build efficient African-accented English ASR models. We introduced the U-WER metric to track model adaptation to intricate accents. Our experiments showed a remarkable 27% WER ratio improvement while reducing the data required for effective training by approximately 45% compared to existing baselines. This demonstrates our approach's efficiency and potential to significantly lower the barriers to ASR technologies in underserved regions. Our method enhances model robustness and generalization across various domains, datasets, and accents, which are crucial for scalable ASR systems. This also helps mitigate bias in ASR technologies, promoting more inclusive and fair AI applications.

## 6 LIMITATIONS

In discussing trade-offs (Section 4), we noted that while our approach enhances performance, particularly with linguistically rich accents, a stopping criterion is essential for complex domains like the **clinical** one to balance adaptation rounds with the pool size. With better resources, we would consider implementing Deep Ensembles (Lakshminarayanan et al. (2017)) as an alternative to our current MC-Dropout method for estimating epistemic uncertainty and leveraging other acquisition functions (such as BALD, BatchBALD) highlighted in this work.

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

# A  APPENDICES

## A.1  HYPER-PARAMETERS

Table 5 shows the hyper-parameter settings used in this study. The top-k value in the table is changed according to the domain used in each of the experiments. For example, when conducting experiments in the general domain, we set the value of top-k to 2k.

| Hyper-parameters | Values |
|---|---|
| attention dropout | 0.1 |
| hidden dropout | 0.1 |
| layer drop | 0.1 |
| train batch size | 16 |
| val batch size | 8 |
| number of epochs | 5 |
| learning rate | 3e-4 |
| maximum audio length | 260000 |
| maximum label length | 260 |
| minimum transcript length | 10 |
| top_k | 2000, 3500, 6500 |
| domains | general, clinical, all |
| active learning rounds | 3 |
| sampling mode | EU-Most, random |
| MC-Dropout round | 10 |

Table 5: Hyper-parameters summary

## A.2  COUNTRY STATISTICS

Table 6 shows the statistics of the countries across the AfriSpeech-200 dataset.

| Country | Clips | Speakers | Duration (seconds) | Duration (hrs) |
|---|---|---|---|---|
| Nigeria | 45875 | 1979 | 512646.88 | 142.40 |
| Kenya | 8304 | 137 | 75195.43 | 20.89 |
| South Africa | 7870 | 223 | 81688.11 | 22.69 |
| Ghana | 2018 | 37 | 18581.13 | 5.16 |
| Botswana | 1391 | 38 | 14249.01 | 3.96 |
| Uganda | 1092 | 26 | 10420.42 | 2.89 |
| Rwanda | 469 | 9 | 5300.99 | 1.47 |
| United States of America | 219 | 5 | 1900.98 | 0.53 |
| Turkey | 66 | 1 | 664.01 | 0.18 |
| Zimbabwe | 63 | 3 | 635.11 | 0.18 |
| Malawi | 60 | 1 | 554.61 | 0.15 |
| Tanzania | 51 | 2 | 645.51 | 0.18 |
| Lesotho | 7 | 1 | 78.40 | 0.02 |

Table 6: Countries Statistics across the dataset

## A.3  DATASET ACCENTS STATS

Tables 7 and 8 provide a list of AfriSpeech accents along with the number of unique speakers, countries where speakers for each accent are located, duration in seconds for each accent, and their presence in the train, dev, and test splits.

## A.4  MOST COMMON ACCENT DISTRIBUTION

Figures 5 and 6 show the most common accent distribution across the general domain with random and EU-Most selection strategies.

| Accent | Clips | Speakers | Duration(s) | Countries | Splits |
|---|---|---|---|---|---|
| yoruba | 15407 | 683 | 161587.55 | US,NG | train,test,dev |
| igbo | 8677 | 374 | 93035.79 | US,NG,ZA | train,test,dev |
| swahili | 6320 | 119 | 55932.82 | KE,TZ,ZA,UG | train,test,dev |
| hausa | 5765 | 248 | 70878.67 | NG | train,test,dev |
| ijaw | 2499 | 105 | 33178.9 | NG | train,test,dev |
| afrikaans | 2048 | 33 | 20586.49 | ZA | train,test,dev |
| idoma | 1877 | 72 | 20463.6 | NG | train,test,dev |
| zulu | 1794 | 52 | 18216.97 | ZA,TR,LS | dev,train,test |
| setswana | 1588 | 39 | 16553.22 | BW,ZA | dev,test,train |
| twi | 1566 | 22 | 14340.12 | GH | test,train,dev |
| isizulu | 1048 | 48 | 10376.09 | ZA | test,train,dev |
| igala | 919 | 31 | 9854.72 | NG | train,test |
| izon | 838 | 47 | 9602.53 | NG | train,dev,test |
| kiswahili | 827 | 6 | 8988.26 | KE | train,test |
| ebira | 757 | 42 | 7752.94 | NG | train,test,dev |
| luganda | 722 | 22 | 6768.19 | UG,BW,KE | test,dev,train |
| urhobo | 646 | 32 | 6685.12 | NG | train,dev,test |
| nembe | 578 | 16 | 6644.72 | NG | train,test,dev |
| ibibio | 570 | 39 | 6489.29 | NG | train,test,dev |
| pidgin | 514 | 20 | 5871.57 | NG | test,train,dev |
| luhya | 508 | 4 | 4497.02 | KE | train,test |
| kinyarwanda | 469 | 9 | 5300.99 | RW | train,test,dev |
| xhosa | 392 | 12 | 4604.84 | ZA | train,dev,test |
| tswana | 387 | 18 | 4148.58 | ZA,BW | train,test,dev |
| esan | 380 | 13 | 4162.63 | NG | train,test,dev |
| alago | 363 | 8 | 3902.09 | NG | train,test |
| tshivenda | 353 | 5 | 3264.77 | ZA | test,train |
| fulani | 312 | 18 | 5084.32 | NG | test,train |
| isoko | 298 | 16 | 4236.88 | NG | train,test,dev |
| akan (fante) | 295 | 9 | 2848.54 | GH | train,dev,test |
| ikwere | 293 | 14 | 3480.43 | NG | test,train,dev |
| sepedi | 275 | 10 | 2751.68 | ZA | dev,test,train |
| efik | 269 | 11 | 2559.32 | NG | test,train,dev |
| edo | 237 | 12 | 1842.32 | NG | train,test,dev |
| luo | 234 | 4 | 2052.25 | UG,KE | test,train,dev |
| kikuyu | 229 | 4 | 1949.62 | KE | train,test,dev |
| bekwarra | 218 | 3 | 2000.46 | NG | train,test |
| isixhosa | 210 | 9 | 2100.28 | ZA | train,dev,test |
| hausa/fulani | 202 | 3 | 2213.53 | NG | test,train |
| epie | 202 | 6 | 2320.21 | NG | train,test |
| isindebele | 198 | 2 | 1759.49 | ZA | train,test |
| venda and xitsonga | 188 | 2 | 2603.75 | ZA | train,test |
| sotho | 182 | 4 | 2082.21 | ZA | dev,test,train |
| akan | 157 | 6 | 1392.47 | GH | test,train |
| nupe | 156 | 9 | 1608.24 | NG | dev,train,test |
| anaang | 153 | 8 | 1532.56 | NG | test,dev |
| english | 151 | 11 | 2445.98 | NG | dev,test |
| afemai | 142 | 2 | 1877.04 | NG | train,test |
| shona | 138 | 8 | 1419.98 | ZA,ZW | test,train,dev |
| eggon | 137 | 5 | 1833.77 | NG | test |
| luganda and kiswahili | 134 | 1 | 1356.93 | UG | train |
| ukwuani | 133 | 7 | 1269.02 | NG | test |
| sesotho | 132 | 10 | 1397.16 | ZA | train,dev,test |
| benin | 124 | 4 | 1457.48 | NG | train,test |
| kagoma | 123 | 1 | 1781.04 | NG | train |
| nasarawa eggon | 120 | 1 | 1039.99 | NG | train |
| tiv | 120 | 14 | 1084.52 | NG | train,test,dev |
| south african english | 119 | 2 | 1643.82 | ZA | train,test |
| borana | 112 | 1 | 1090.71 | KE | train |

Table 7: Dataset Accent Stats, Part I

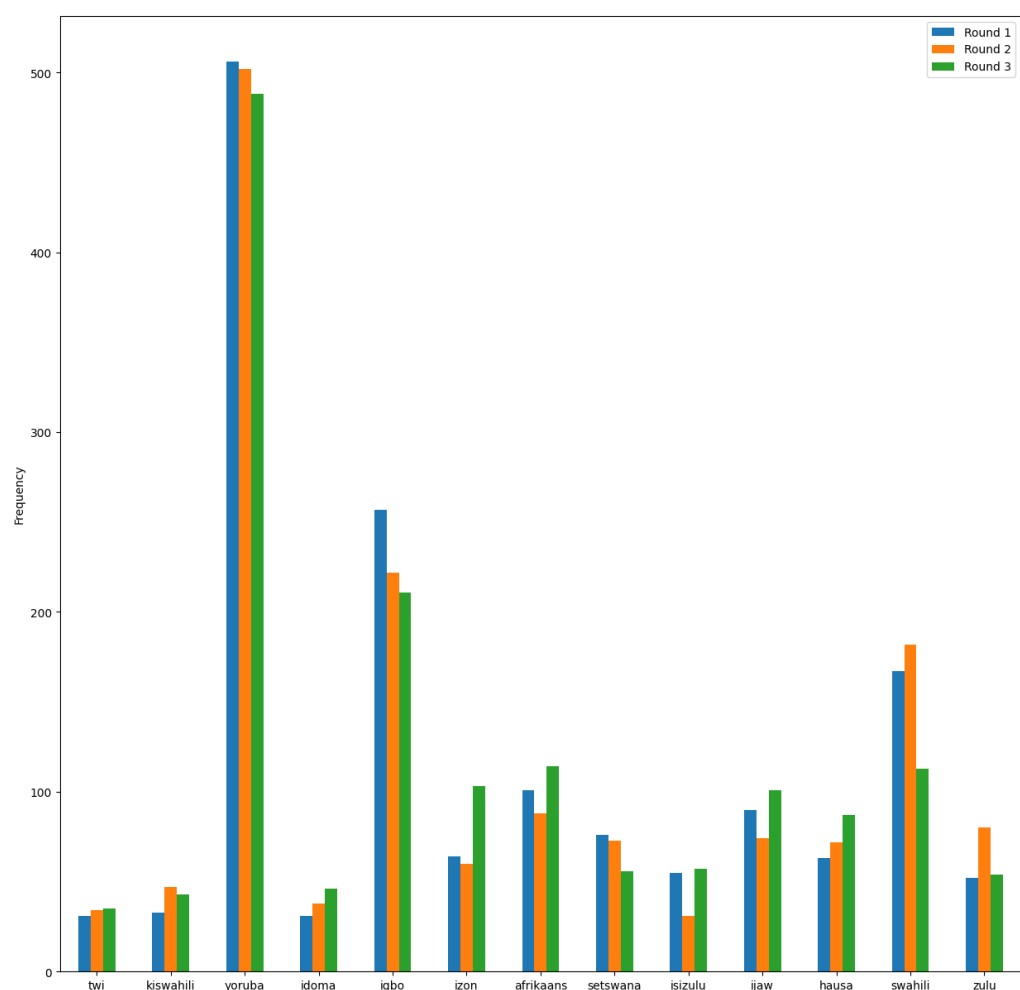

Figure 5: Most common accents distribution across the general domain with EU-Most sampling strategy.

## A.5    ASCENDING AND DESCENDING ACCENTS

Figure 7 shows ascending and descending accents across the Top 2k *most* uncertain samples.

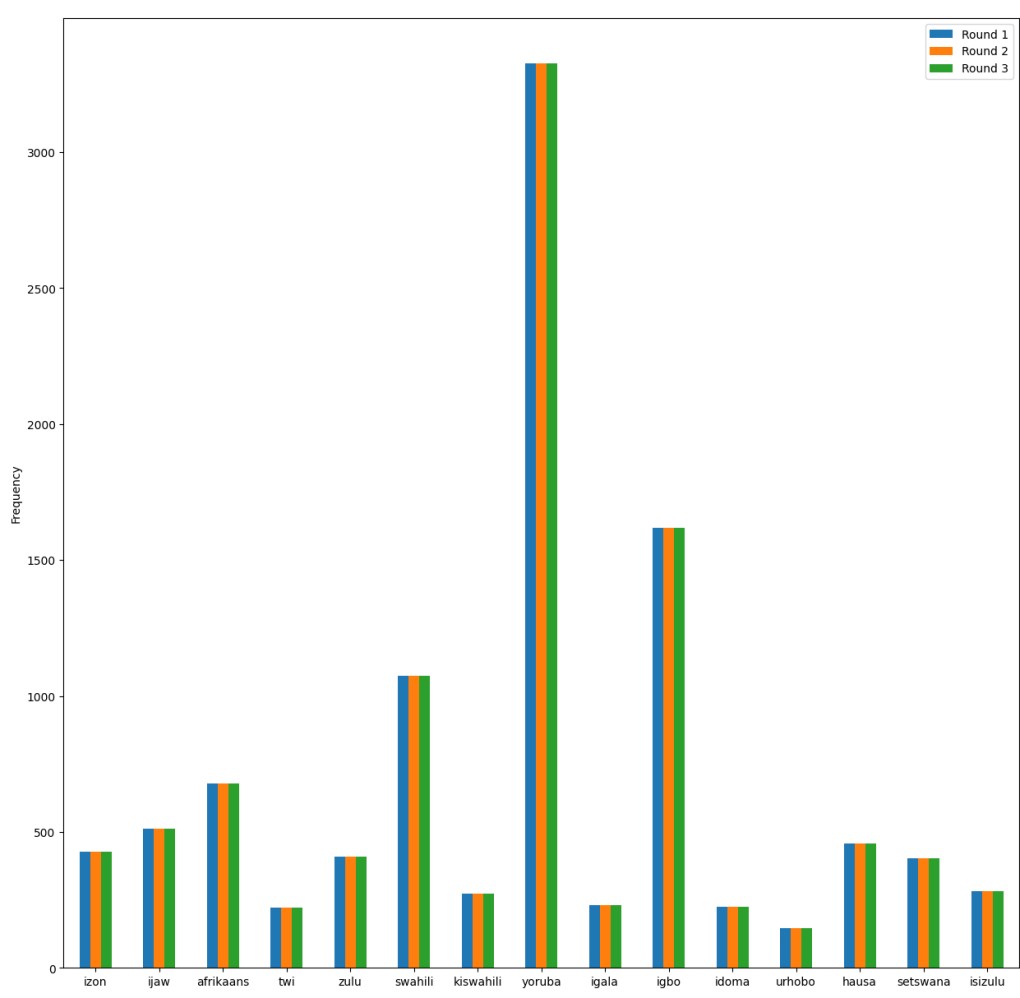

Figure 6: Most common accents distribution across the general domain with random selection strategy.

| Accent | Clips | Speakers | Duration(s) | Countries | Splits |
|---|---|---|---|---|---|
| swahili ,luganda ,arabic | 109 | 1 | 929.46 | UG | train |
| ogoni | 109 | 4 | 1629.7 | NG | train,test |
| mada | 109 | 2 | 1786.26 | NG | test |
| bette | 106 | 4 | 930.16 | NG | train,test |
| berom | 105 | 4 | 1272.99 | NG | dev,test |
| bini | 104 | 4 | 1499.75 | NG | test |
| ngas | 102 | 3 | 1234.16 | NG | train,test |
| etsako | 101 | 4 | 1074.53 | NG | train,test |
| okrika | 100 | 3 | 1887.47 | NG | train,test |
| venda | 99 | 2 | 938.14 | ZA | train,test |
| siswati | 96 | 5 | 1367.45 | ZA | dev,train,test |
| damara | 92 | 1 | 674.43 | NG | train |
| yoruba, hausa | 89 | 5 | 928.98 | NG | test |
| southern sotho | 89 | 1 | 889.73 | ZA | train |
| kanuri | 86 | 7 | 1936.78 | NG | test,dev |
| itsekiri | 82 | 3 | 778.47 | NG | test,dev |
| ekpeye | 80 | 2 | 922.88 | NG | test |
| mwaghavul | 78 | 2 | 738.02 | NG | test |
| bajju | 72 | 2 | 758.16 | NG | test |
| luo, swahili | 71 | 1 | 616.57 | KE | train |
| dholuo | 70 | 1 | 669.07 | KE | train |
| ekene | 68 | 1 | 839.31 | NG | test |
| jaba | 65 | 2 | 540.66 | NG | test |
| ika | 65 | 4 | 576.56 | NG | test,dev |
| angas | 65 | 1 | 589.99 | NG | test |
| ateso | 63 | 1 | 624.28 | UG | train |
| brass | 62 | 2 | 900.04 | NG | test |
| ikulu | 61 | 1 | 313.2 | NG | test |
| eleme | 60 | 2 | 1207.92 | NG | test |
| chichewa | 60 | 1 | 554.61 | MW | train |
| oklo | 58 | 1 | 871.37 | NG | test |
| meru | 58 | 2 | 865.07 | KE | train,test |
| agatu | 55 | 1 | 369.11 | NG | test |
| okirika | 54 | 1 | 792.65 | NG | test |
| igarra | 54 | 1 | 562.12 | NG | test |
| ijaw(nembe) | 54 | 2 | 537.56 | NG | test |
| khana | 51 | 2 | 497.42 | NG | test |
| ogbia | 51 | 4 | 461.15 | NG | test,dev |
| gbagyi | 51 | 4 | 693.43 | NG | test |
| portuguese | 50 | 1 | 525.02 | ZA | train |
| delta | 49 | 2 | 425.76 | NG | test |
| bassa | 49 | 1 | 646.13 | NG | test |
| etche | 49 | 1 | 637.48 | NG | test |
| kubi | 46 | 1 | 495.21 | NG | test |
| jukun | 44 | 2 | 362.12 | NG | test |
| igbo and yoruba | 43 | 2 | 466.98 | NG | test |
| urobo | 43 | 3 | 573.14 | NG | test |
| kalabari | 42 | 5 | 305.49 | NG | test |
| ibani | 42 | 1 | 322.34 | NG | test |
| obolo | 37 | 1 | 204.79 | NG | test |
| idah | 34 | 1 | 533.5 | NG | test |
| bassa-nge/nupe | 31 | 3 | 267.42 | NG | test,dev |
| yala mbembe | 29 | 1 | 237.27 | NG | test |
| eket | 28 | 1 | 238.85 | NG | test |
| afo | 26 | 1 | 171.15 | NG | test |
| ebiobo | 25 | 1 | 226.27 | NG | test |
| nyandang | 25 | 1 | 230.41 | NG | test |
| ishan | 23 | 1 | 194.12 | NG | test |
| bagi | 20 | 1 | 284.54 | NG | test |
| estako | 20 | 1 | 480.78 | NG | test |
| gerawa | 13 | 1 | 342.15 | NG | test |

Table 8: Dataset Accent Stats, Part II

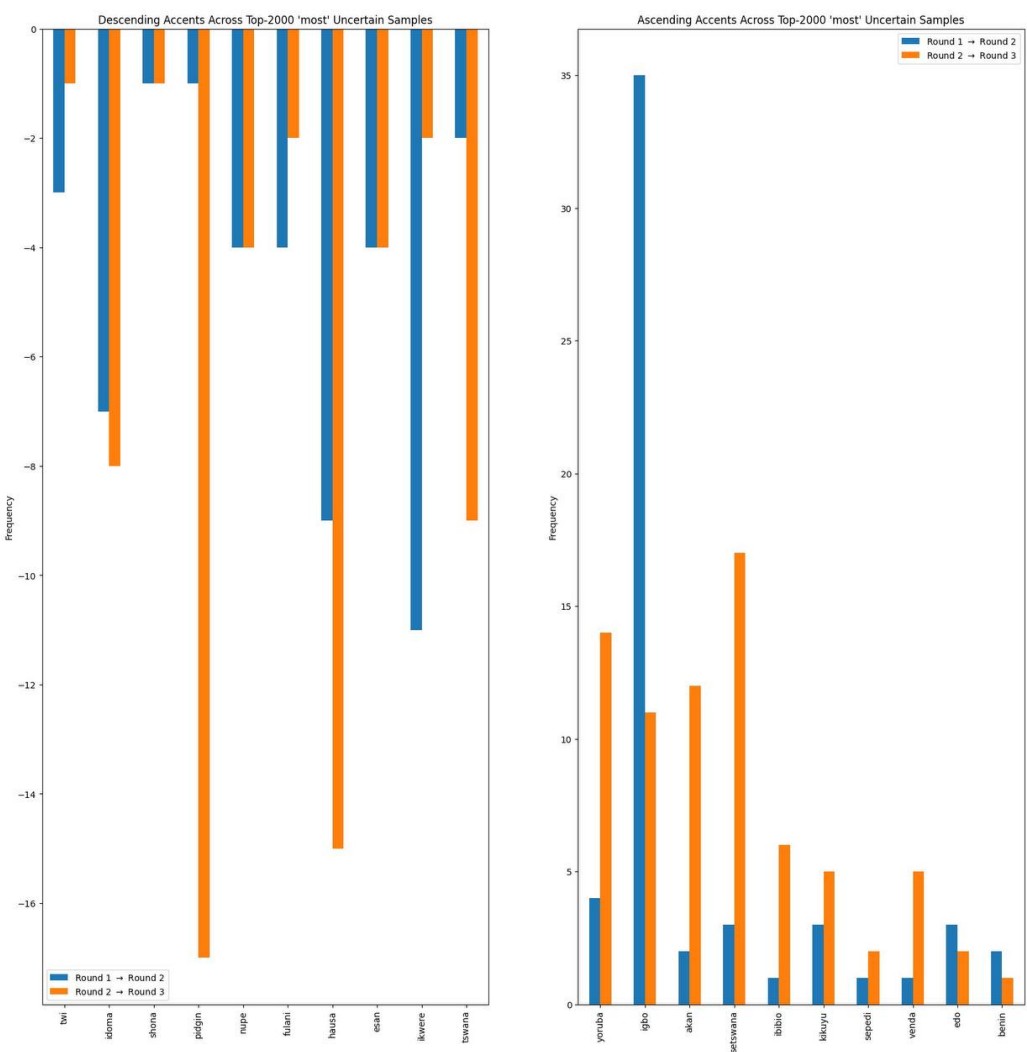

Figure 7: Ascending and descending accents across Top-2K *most* uncertain samples.

