# OpenReview forum: "Advancing African-Accented English Speech Recognition: Epistemic Uncertainty-Driven Data Selection for Generalizable ASR Models"
_ICLR.cc/2025/Conference — ICLR 2025 Conference Withdrawn Submission_

### Official Review · Reviewer_q5Py · 2024-10-28

**Soundness:** 3
**Presentation:** 2
**Contribution:** 3
**Rating:** 3
**Confidence:** 3

**Summary:**

This paper combines active learning and Monte Carlo dropout based uncertainty filtering in order to adapt ASR models for different datasets, in this case mainly to unseen accents. The training algorithm splits the existing dataset into two sets, and tunes the model on the larger set, then uses uncertainty sampling to sample the most ambiguous data samples from the other set. Finally, the first set and the selected samples are using in a second finetuning step. Uncertainty is determined by the variance of the WERs on masked inputs during training iterations. The paper calls this measure U-WER and the paper also argues that it introduces this metric to judge the adaptation perfromance.
Experiments on various African-accented datasets, and also some medical speech dataset investigate whether the proposed selection method outperforms random sampling and whether the approach is applicable to different ASR models. The uncertainty-based selection approach outperforms the baselines across all models, domains, and datasets reaching up to 27% relative WER reduction with smaller number of training samples.

**Strengths:**

Originality,
- Active learning for ASR is relatively under-investigated although there are several data selection studies for ASR. The paper investigates a Monte Carlo dropout based uncertainty criterion to sample the uncertain data points which is novel enough.

Quality,
- The paper demonstrates that the results apply to various ASR models which is an important feature.

Clarity,
- Text is mostly clear. Even though there are some issues (please check the weaknesses section, the second item)

Significance
- Achieving WER reduction on low-resource domains is still an important problem and by providing a sampling method reducing the data requirement, the paper remains relevant to the ASR community.

**Weaknesses:**

1. Even though the paper's main proposal is using active learning with uncertainty sampling, the results suggest that adding the selected samples to the original training data improves the WER more than finetuning on selected samples (numerical results in Table 1). Both approaches result in improvements but the main improvement is not coming from the pipeline described in Fig. 1. Section 4 also mainly discusses the EU-most approach, hence the AL + EU-most results can be deferred to the Appendix.

2. (To the best of my understanding) It seems that Figs. 2-4 are not very well labeled and I do not see a clear explanation of what the three separate figures mean. Please check.

**Questions:**

1. Do the authors have any additional comments on the smaller improvements on the clinical domain ASR? Could it be due to the change in vocabulary as compared to the general ASR domain?

2. Please makes Figs. 2-4 more clearer.

3. [Minor] Text font size in tables are quite small, please check your formatting options.

4. The literature review in Section 2 refers to a very limited number of studies, it could have been useful to include some ASR papers that consider data sampling approaches for domain adaptation or finetuning. There are various studies using pretrained ASR confidence scores to select data samples in earlier studies. Please check.

---

### Official Review · Reviewer_gNRX · 2024-11-02

**Soundness:** 2
**Presentation:** 2
**Contribution:** 3
**Rating:** 3
**Confidence:** 4

**Summary:**

This paper focuses on improving African-accented ASR via an Active Learning process. The authors considered a setting where they have access to labeled training data to fine-tune a pre-trained ASR, and an unlabeled dataset from which top accent-wise uncertain samples are selected and added to the ASR training data. For uncertainty estimation, the authors proposed an algorithm to sample multiple sub-model outputs from a model trained with Dropout, based on which the WER variance indicates uncertainty. The empirical results show that the AL process with the proposed data selection strategies has improved the ASR performance on multiple datasets and multiple accents.

**Strengths:**

This paper contributed to improving African-accented ASR. This is an important advancement towards fair and inclusive technologies for the African community.  This work investigated multiple African-accented speech corpora and experimented with several strong SSL models, Conformer and Whisper, which is nice.

**Weaknesses:**

Overall, I think the presentation/description of the experimental setups should be improved. In Figures 2-4, the y-axis labels states U-WER, which is essentially a standard deviation metric measuring uncertainty. However, the captions states "WER" which indicates model final performance. These two are different metrics, which is confusing to me. Figures 2-4 are also hard to read: (1) the red and blue boxes should be vertically aligned (also, why the blue boxes on the right hand side have the bizarre shapes?); (2) subplots should be aligned as well; (3) the legend should be "EU-Most" not "Most" for consistency; (3) the resolution of Figures 2-4 should be improved.

Also, Table 4 experimental setting is not shown. Which model is this? In general,  I think the paper lacks experimental setup details which causes some confusion - see my question in the next section.

**Questions:**

1. Section 3.3 - What is the difference between "EU-Most" and "AL-EU-Most" strategies? My understanding is that AL is also needed for "EU-Most"?
2. Table 3 - What is the training data for "Baseline"? Section 3.3 states that "Baseline" is trained using $D_{train}$. Is it only 30% of the original data, or the entire size of the original data? Is it a standard fine-tuning of the pre-trained models? In general, it will be better to clearly describe all data used for each model, including the baseline, so that readers can understand the comparison condition.
3. Do you train accent-specific models? The U-WER metric is accent-dependent. So, do you pool all accent-wise selected data together and merge them, or do you fine-tune accent-specific models?
4. Figures 2-4 - as I said in the Weaknesses section above, U-WER is not WER. Here, to demonstrate the performance improvement, accent-wise WER should be shown.
5. For AL, do you perform continual training or re-train the base model using the pooled selected data? Again, this should be made clearer in the experimental setting.
6. It would be informative to show the uncertainty change across different rounds. Is the final resulting model more accent-wise certain or uncertain?

---

### Official Review · Reviewer_TEax · 2024-11-03

**Soundness:** 3
**Presentation:** 3
**Contribution:** 2
**Rating:** 5
**Confidence:** 4

**Summary:**

This paper presents experiments in active learning and data selection for training ASR on African-Accented English.  The central techniques are sound, and the results are effective.  However, the contributions are stronger in their application of techniques rather than the specific contributions to active learning.

**Strengths:**

The introductory material demonstrates a clear positioning of the different techniques available for active learning, uncertainty sampling and the relative strengths of these techniques.

The paper is generally quite easy to read.

The results demonstrate the value of U-WER as a selection criteria, however the AL variant does not improve over EU-Most selection strategy.

**Weaknesses:**

While the paper spends a good amount of motivating material on positioning Active Learning as valuable here, the results suggest a more naive approach (EU-Most) to be more effective, at least for this task.

U-WER is calculated as the standard deviation of WER over MC-dropout augmentation.  Were other augmentations explored for this estimation? How important is it that Dropout was applied during pre-training of the ASR models used for this finding to hold?

**Questions:**

The data set focuses on English spoken by speakers from the US and sub-Saharan Africa.  I'm curious if this suggests a more precise title than "Advancing Africa-Accented English", since northern Africa is omitted from this study.

Is there anything specific to the English spoken by African speakers that would suggest that the result would (or would not) transfer to other English variants (regional, language learners from countries that are not anglophone, etc.)? Would the authors expect this technique to be similarly useful in improving robustness of other languages to their accents?

Section 3.2.1 claims that "Ideally U-WER->0".  Is this correct? This would suggest that the underlying model does not change its prediction based on dropout.  While robustness is a desirable quality, one of the strengths of dropout is diversification of behavior under different dropout settings, enabling a more robust final model.  (e.g. https://arxiv.org/abs/1207.0580, https://arxiv.org/abs/1706.06859,  To what extent does the U-WER stabilize, and is convergence to zero attained?

---

### Note · Authors · 2024-11-12

I have read and agree with the venue's withdrawal policy on behalf of myself and my co-authors.